# Protective Effect of *Curculigo orchioides* Gaertn. Extract on Heat Stress-Induced Spermatogenesis Complications in Murine Model

**Thanh-Nhan Bui-Le** [1,2], **Quang Hoang-Tan** [3], **Huong Hoang-Viet** [4], **Bich-Phuong Truong-Thi** [2] **and Tung Nguyen-Thanh** [1,5,*]

1    Faculty of Basic Science, University of Medicine and Pharmacy, Hue University, Hue 49000, Vietnam
2    Faculty of Biology, University of Sciences, Hue University, Hue 49000, Vietnam
3    Institute of Biotechnology, Hue University, Hue 49000, Vietnam
4    Thua Thien Hue Department of Science and Technology, Hue 49000, Vietnam
5    Institute of Biomedicine, University of Medicine and Pharmacy, Hue University, Hue 49000, Vietnam
*    Correspondence: nguyenthanhtung@hueuni.edu.vn or nttung@huemed-univ.edu.vn

**Abstract:** *Curculigo orchioides* Gaertn. is a precious herb used in traditional medicine systems in Asian countries for various health benefits. This study investigated the potential protective effects of *C. orchioides* extract on reproductive health under heat stress conditions in male mice. Forty-eight mice were divided into eight groups, control condition (C group), *C. orchioides* extract at the dosages of 100, 200, and 400 mg/kg/day (C100, C200, C400 group), 40 °C heat exposure (H group), and combined 40 °C heat exposure and *C. orchioides* extract at the dosages of 100, 200, and 400 mg/kg/day (HC100, HC200, HC400 group). The result shows that the mice that received only *C. orchioides* extract without heat stress do not have a significant change in histological structure and testosterone level. The histological analysis of testicular tissue showed that heat stress conditions reduced reproductive function and inhibited the spermatogenesis of male mice. The *C. orchioides* rhizome extract treatment attenuated the heat stress-induced spermatogenesis complications in the murine model. Mice in the heat-stress group treated with *C. orchioides* extract had increased spermatogenic cells and spermatozoa compared with mice exposed to heat without *C. orchioides* treatment. Moreover, the aqueous extract of *C. orchioides* rhizome enhanced the serum total testosterone levels in heat-exposed mice. In conclusion, the study findings validate that *C. orchioides* is effective against heat stress-induced spermatogenesis complications in the murine model.

**Keywords:** heat stress; spermatogenesis; *Curculigo orchioides*; lycorine

## 1. Introduction

*Curculigo orchioides* Gaertn. (*C. orchioides*), the widely known medicinal herb, belongs to the family Hypoxidaceae, referred to as Kalimusali, Golden Eye Grass, Nilappanai, and Sam cau, occurring in many tropical areas (China, India, Vietnam et al.) [1–5]. *C. orchioides* is a herbaceous perennial with a cylindrical rhizocorm, numerous lateral roots, an elongate rootstock, and a light aroma [4,6]. The main phytoconstituents of *C. orchioides* are glycosides, alkaloids, triterpenoids, saponins, flavones, and polysaccharides, which were isolated from the rhizome or the whole plant [7–10]. According to Pham Thanh et al., *C. orchioides* collected in the center of Vietnam contain high orcinol glucoside, a major phenolic glucoside compound with many medicinal properties [11].

*C. orchioides* has been widely used in Asian indigenous medicine and has suggested hepatoprotective, antioxidant, anti-cancer [12,13], neuroprotective, and anti-inflammatory affects [12,14]. *C. orchioides* has also been used to enhance memory [15], improve the immune system and rejuvenation [5,16], and to treat a variety of diseases including diabetes [12,13]. The rhizome extract is also used as a tonic for overcoming impotence [17], as

a sedative, a neuroprotective, and for anti-necrosis [18,19] and antibacterial activity [20]. *C. orchioides* has great potential for development as an effective drug for the treatment of many diseases. Yongbo Yu et al. purified the active substances in the crude polysaccharide from *C. orchioides* and show that it significantly enhanced the mineralization of the osteoblasts [21]. *C. orchioides* is considered a sexual tonic in traditional medicine. Chauhan et al. showed that an ethanolic extract of *C. orchioides* rhizomes had a pronounced effect on the orientation of male rats towards female rats, as evidenced by the increasingly frequent and vigorous anogenital sniffing and mounting. *C. orchioides* increased spermatogenesis via histoarchitecture changes and an increase in the number of spermatocytes and spermatids [22].

There are many factors that cause a decline in male reproductive health and semen quality [23]. Heat stress is one of the risk factors that affects most male reproductive functions in mammals. High scrotal temperature is a significant cause of male factor infertility. There are many common thermogenic factors that affect scrotal temperature, including lifestyle and behavioral influences, occupational and environmental factors, and pathophysiological causes [24]. Spermatogenesis is a temperature-dependent process, and any increase above the normal scrotal temperature range can result in the death of heat-vulnerable germ cells and disrupted somatic cell function [25].

Recently, use of traditional medicines from medicinal plants has increased in treating male infertility. Many studies show evidence of the effect of medicinal herbs on the testicular tissue, epididymis, spermatogenesis, steroidogenesis, testosterone production, erection function, ejaculation, and libido [26,27]. Therefore, the aim of this study was to investigate the effect of *Curculigo orchioides* extract on testicular histopathology alteration and blood testosterone levels in heat stress-induced male mice.

## 2. Materials and Methods

### 2.1. Plant Collection and Extraction

One and a half kilogram of rhizomes of *C. orchioides* (around 150 plants) were harvested in Thua Thien Hue province, Vietnam. Plants were cleaned and dried at 45 °C in the incubator. Three hundred and thirty grams of dried herbal was pulverized. (Figure 1). The herbal powder was then suspended in deionized water and extracted by placing it in a capped glass jar over a magnetic stirrer at 4 °C for 10 min, then ultrasonicated (S10H, Elmasonic) at 40 Hz for 30 min. The extract was subsequently filtered using a 0.45 μm Minissart filter (Sartorius, Germany). The extract was concentrated at 45 °C to obtain a brownish mass and stored in a refrigerator at 4 °C, then diluted with distilled water in different doses to be given to mice.

### 2.2. Qualitative and Quantitative Determination of Alkaloids

The qualitative estimation of alkaloids was made using Mayer's reagents (catalog No. DD-056; ABT, Ha Noi City, Vietnam), Dragendorff's reagents (catalog No. DD-055; ABT, Ha Noi City, Vietnam), and Bouchardat's reagents (catalog No. DD-057; ABT, Ha Noi City, Vietnam). In Mayer's test, 1 mL of plant extract was treated with 3–4 drops of Mayer's reagent and examined for the formation of a yellowish cream precipitate. The appearance of a green-colored precipitate indicated the presence of alkaloids. In Dragendorff's test, 3–4 drops of Dragendorff reagent were added to 1 mL of the extract. The appearance of yellow-orange to red-orange colored precipitate indicated the presence of alkaloids. In Bouchardat's test, 1 mL of the extract and 3–4 of Bouchard at drops were added, and a reddish brown crystalline precipitate indicates the presence of alkaloid [28,29]. To determine the alkaloid content of different samples, some part of the extract residue was dissolved in 2N HCl and filtered. We transferred 1 mL of this solution to a separating funnel and washed it 3 times with 10 mL chloroform. The pH of this solution was adjusted to neutral by 0.1N NaOH. After that, 5 mL of BCG solution and 5 mL of phosphate buffer were added to this solution. The complex was extracted with 1, 2, 3, and 4 mL chloroform by vigorous shaking, the extracts were collected in a 10 mL volumetric flask

and diluted with chloroform up to the mark. Subsequently, quantification was conducted spectrophotometrically [29].

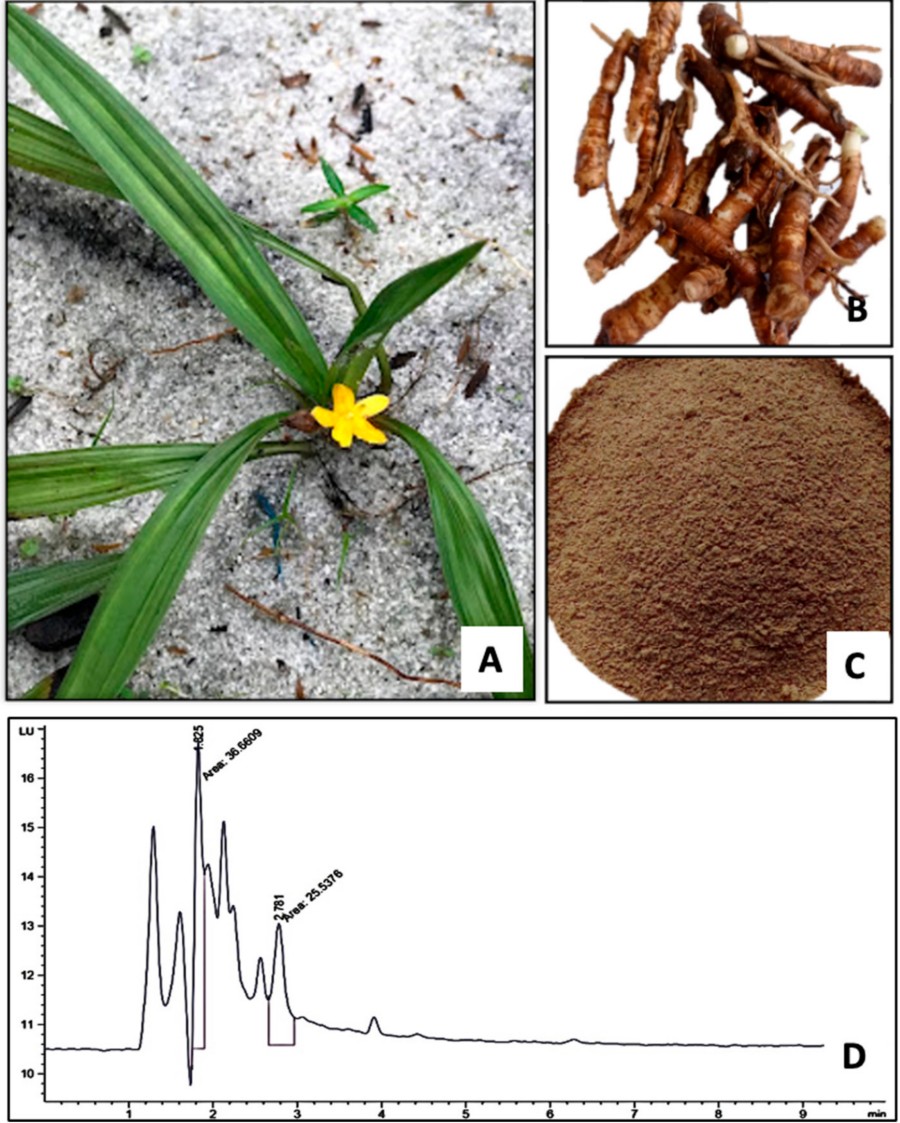

**Figure 1.** The characteristics of *Curculigo orchioides* Gaertn. (**A**)The whole plant in nature; (**B**) the rhizomes; (**C**) the herbal powder; (**D**) HPLC quantification of Lycorine in Curculigo orchioides.

*2.3. High-Pressure Liquid Chromatographic Analysis of Lycorine*

Aliquots of the extracts of *C. orchioides* were were diluted 20-fold with deionized water and injected into the HPLC system. HPLC analysis was carried out using an Accela HPLC liquid phase system (Thermo Fisher Scientific, Waltham, MA, USA). An Intersil $C_{18}$ column (5 mm, 4.6 mm 250 mm; GL Sciences Inc., Tokyo, Japan) was used as the stationary phase. The mobile phase was a mixture of $KH_2PO_4$ buffer (15 mM, pH 6.35) with methanol (50:50, *v/v*). The mobile phase was filtered through a 0.45 mm nylon filter membrane and degassed in an ultrasonic bath for 10 min before use. The temperature of the column oven was set at 30 °C, and the flow rate was 0.8 mL/min. The injection volume was 20 mL [30]. The standard lycorine was purchased from Sigma-Aldrich (St. Louis, MO, USA). Analytical grade methanol was used for the extraction and HPLC grade acetonitrile and methanol used for analysis. Both were purchased from Merck (Darmstadt, Germany).

### 2.4. Animals

Adult male Swiss mice weighing 20–23 g (8–12 weeks old) purchased from Pasteur Institute of Nha Trang (Viet Nam) were fed on a standard pellet diet and water ad libitum. For the current study, 48 male mice who had never experienced sexual behavior before were used. The animals were kept at room temperature (26 ± 2C) on a 12-h light/12-h dark photoperiod. All the animals were allowed to acclimatize in the test cage 7 days prior to experimentation. All animal experimentation studies were out carried after prior permission from the Animal Ethics Committee of Hue University, Vietnam (Certificate no. HU VN0007).

The animals were randomly divided into 8 groups, each group consisting of 6 mice: (C) Control group and (H) 40 °C heat exposure groups were both given plain distilled water to drink; (C100) *C. orchioides* extract 100 mg/kg/day group; (C200) 200 mg/kg/day *C. orchioides* extract group; (C400) 400 mg/kg/day *C. orchioides* extract group; (HC100) 100 mg/kg/day *C. orchioides* extract + heat stress group; (HC200) 200 mg/kg/day *C. orchioides* extract + heat stress group; (HC400) 400 mg/kg/day *C. orchioides* extract + heat stress group. Aqueous extracts were orally suspended using 20-gauge stainless steel feeding needles.

### 2.5. Mouse Model for Testicular Heat Stress

For five weeks, the lower half of the mouse's body, including the scrotum, was immersed in a temperature-controlled water bath (40 °C) using a keeper tool for 10 min twice per day at 10-min intervals. Control mice were treated in the same way, but in a water bath maintained at 26 °C. After the bath, the mice were dried and examined for any injury or redness to the scrotal skin before being returned to their cages. The mice have been monitored for their general health. After completing the heat exposure experiment for 5 weeks, mice were anesthetized with 80 mg/kg ketamine and 16 mg/kg xylazine and then cardiac puncture for blood collection was performed as a terminal procedure. Their testis tissue was harvested for histological morphometric analysis [31].

### 2.6. Total Testosterone Hormone Measurements

Mice were anesthetized and then we collected the circulation blood by cardiac puncture. The blood serum concentration of total testosterone was measured by electrochemiluminescence immunoassay using Elecsys Testosterone II (05200067190, Roche, Basel, Switzerland) from Roche Diagnostics, according to the manufacturer's instructions, with the cobas® 8000 modular (Roche Diagnostics, Mannheim, Germany). The sensitivities of hormone detected per assay tube were 0.025 ng/mL.

### 2.7. Hematoxylin-Eosin Staining and Histopathology Assessment

Testicular specimens were individually immersed in 4% buffered formaldehyde for 24 h, dehydrated with graded concentrations of ethyl alcohols, and tissues were embedded into paraffin blocks. Paraffin sections 5 μm thick were obtained by microtome (Leica RM2255, Germany) and transferred onto gelatin-coated slides, deparaffinized with xylene, and then rehydrated through a descending series of ethanol and distilled water. Slides were stained with hematoxylin and eosin (H&E) (Sigma, Vienna, Austria) and observed under a light microscope for histopathological analysis.

The testicular tissue was evaluated in random order using Zeiss Axioplan-2 Imaging light microscopy (KS-300 Imaging System). Testicular damage and spermatogenesis were assessed histopathologically using Johnsen's mean testicular biopsy score (Table 1) [32]. By using a 40× magnification, forty-five tubules for each animal were graded. Testicular biopsies were blinded and given a score according to the following Johnsen scoring system by two independent authors.

**Table 1.** Histological classification of seminiferous tubular cross-sections according to the Johnsen scoring system.

| Score | Description |
|:---:|:---|
| 10 | Complete spermatogenesis with many spermatozoa. Germinal epithelium organized in a regular thickness leaving an open lumen. |
| 9 | Many spermatozoa present but germinal epithelium disorganized with marked sloughing or obliteration of lumen. |
| 8 | Only a few spermatozoa (<5–10) present in the section. |
| 7 | No spermatozoa but many spermatids present. |
| 6 | No spermatozoa and only a few spermatids (<5–10) present. |
| 5 | No spermatozoa, no spermatids but several or many spermatocytes present. |
| 4 | Only few spermatocytes (<5) and no spermatids or spermatozoa present. |
| 3 | Spermatogonia are the only germ cells present. |
| 2 | No germ cells but Sertoli cells present. |
| 1 | No cells in the tubular section. |

*2.8. Statistical Analysis*

Results were analyzed using SPSS version 18. All data are expressed as mean $\pm$ SD. The significant differences among the means of the treated groups were analyzed using a Student's *t*-test. A *p*-value lower than 0.05 ($p < 0.05$) was considered to be statistically significant.

**3. Results**

*3.1. Phytochemical Characteristics of Curculigo orchioides Gaertn*

The presence of alkaloids in *C. orchioides* extracts was evidenced by the cream-colored precipitate from the reaction of alkaloids with Mayer's reagent, or orange-colored precipitate with Dragendorff's reagents, and the brown precipitates from Bouchardat's test. The regression was calculated using the formula y = ax + b. The equation obtained from the standard curve is y = 0.3871x + 0.0001, where x is content (mg/mL) and y is absorbance. The *r*-value, calculated from the standard curve, is 0.9997. The total alkaloid content recorded in the extract of *C. orchioides* rhizomes was 2.008 $\pm$ 0.039 mg/mL.

The calibration plots for lycorine were found to be linear within the range of 0.1 to 1.0 µg/mL. The regression equation was y = 159.67x + 3.9613 ($R^2 = 0.9998$). The lycorine content determined by the HPLC system was 2.245 $\pm$ 0.007 (ppm) (Figure 1D).

*3.2. Curculigo orchioides Extract Ameliorates the Heat Stress-Induced Altered Testosterone Concentration Levels*

Quantitative analysis of the testosterone levels in the blood of mice subjected to heat stress with or without *C. orchioides* extract treatment after 6 weeks is presented in Table 2 and Figure 2.

**Table 2.** Effect of *Curculigo orchioides* extract on blood testosterone levels of heat stress-induced male mice.

| Groups | Parameters | Testosterone (nmol/L) |
|:---:|:---:|:---:|
| C | Distilled water | 11.320 $\pm$ 6.192 |
| C100 | 100 mg/kg | 11.401 $\pm$ 2.956 |
| C200 | 200 mg/kg | 11.786 $\pm$ 3.372 |
| C400 | 400 mg/kg | 12.419 $\pm$ 2.618 |
| H | 40 °C + Distilled water | 0.196 $\pm$ 0.154 |
| HC100 | 40 °C + 100 mg/kg | 0.434 $\pm$ 0.487 |
| HC200 | 40 °C + 200 mg/kg | 1.303 $\pm$ 0.917 |
| HC400 | 40 °C + 400 mg/kg | 2.885 $\pm$ 2.021 |

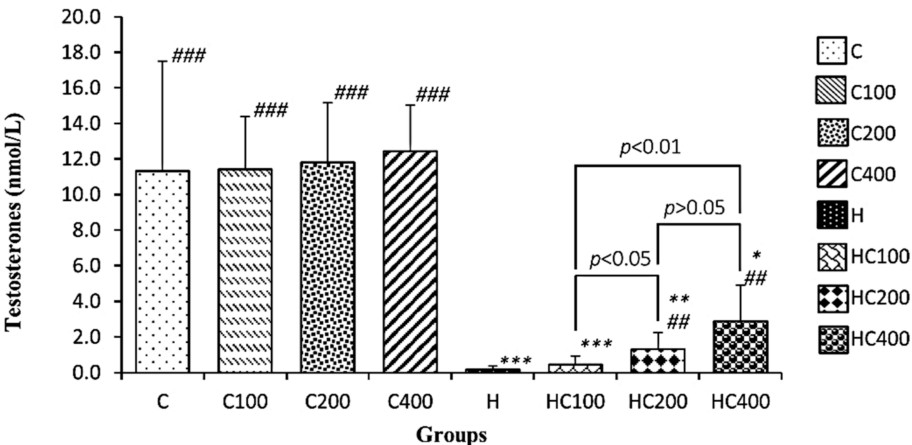

**Figure 2.** Testosterone concentrations in the blood of male Swiss mice. * Compared to the control group (C), # Compared to the heat stress groups, *, #: $p < 0.05$, **, ##: $p < 0.01$, ***, ###: $p < 0.005$.

The result shows that there was a negative effect of heat stress on testosterone concentration in male mice. The value of testosterone concentration in the heat exposure group mice was $0.196 \pm 0.154$ nmol/L, which is significantly lower ($p < 0.005$) compared with those in the control group ($11.320 \pm 6.192$ nmol/L).

There was a slight increase in blood testosterone in the C. orchioides extract 100 mg/kg/day (C100), 200 mg/kg/day (C200), and 400 mg/kg/day (C400) groups compared to the control group ($11.401 \pm 2.956$ nmol/L, $11.786 \pm 3.372$ nmol/L, and $12.419 \pm 2.618$ nmol/L, respectively), but it did not reach statistical significance ($p > 0.05$).

Heat exposure treatment combined with *C. orchioides* extract in 100 mg/kg/day (HC100), 200 mg/kg/day (HC200) and 400 mg/kg/day (HC400) groups increased blood testosterone levels compared to the heat exposure-only group. The testosterone level in the heat exposure treatment combined with 100 mg/kg/day extract ($0.434 \pm 0.487$ nmol/L) increased compared to those in the heat exposure-only group but did not reach a significant difference. Meanwhile, treatment with 200 and 400 mg/kg/day extract significantly increase the testosterone levels in heat-exposure mice ($1.303 \pm 0.917$ nmol/L and $2.885 \pm 2.021$ nmol/L, respectively). *C. orchioides* extract significantly increases testosterone levels in heat-exposure mice. However, their level is still significantly lower compared to those in the control group ($11.320 \pm 6.192$ nmol/L).

### 3.3. Curculigo orchioides Extract Ameliorates Attenuation of Heat Stress-Induced Spermatogenesis Complications

The histopathology of the semi-coniferous tubules of the eight studied groups of mice is illustrated in Figure 3. The result of the light microscopic study showed the normal histoarchitecture of the testes and regular spermatogenesis events in mice without heat stress (C group, C100 group, C200 group, and C400 group) (Figure 3A–D). The germinal epithelium (including cells at different stages of spermatogenic development), was located in invaginations or in dilations between Sertoli cells (Figure 3A′–D′).

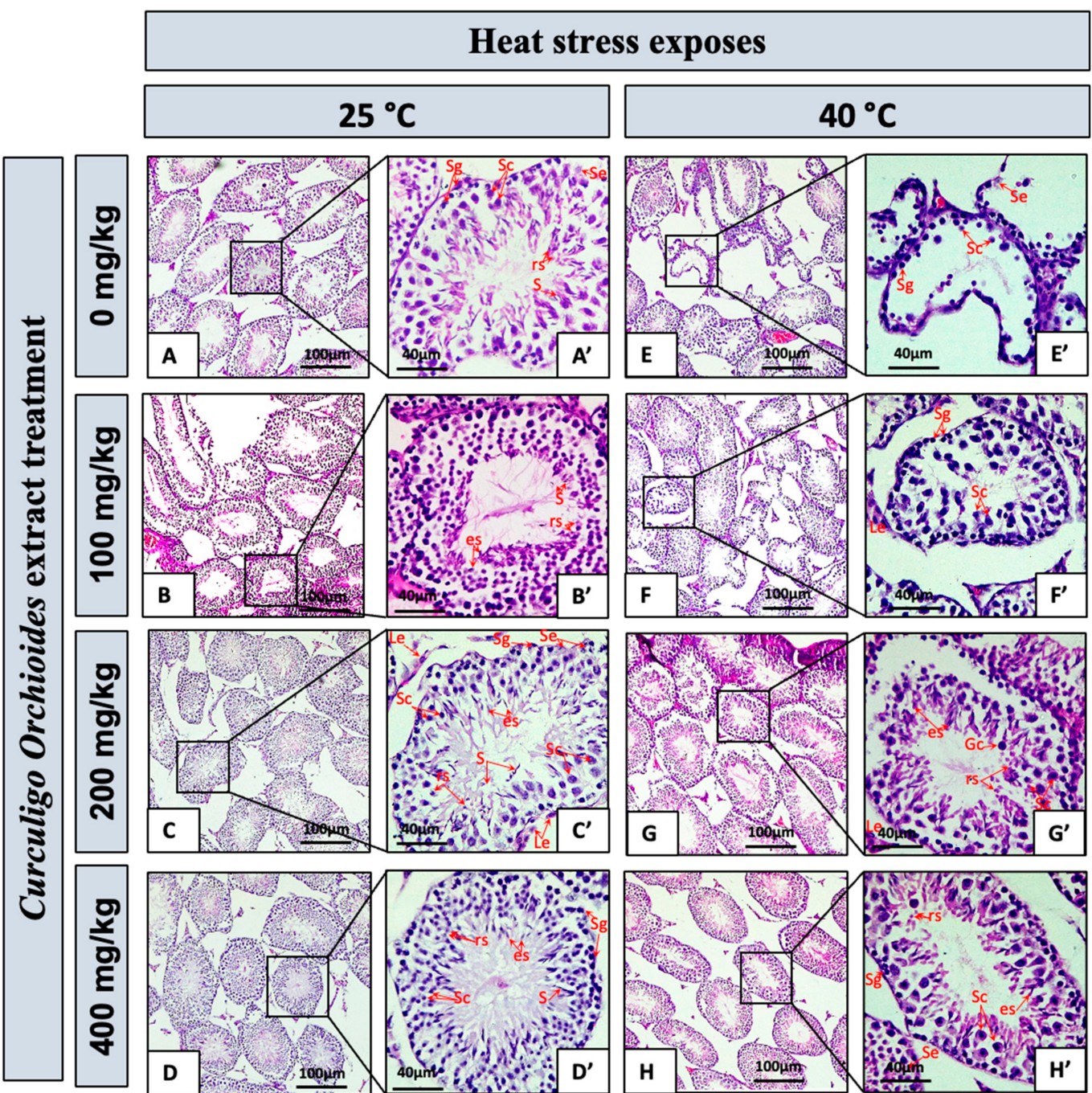

**Figure 3.** Testicular histopathology in mice subjected to heat exposure and treated with *C. orchioides* extract. (**A**,**A′**) Control group; (**B**,**B′**) *C. orchioides* 100 mg/kg bw; (**C**,**C′**) *C. orchioides* 200 mg/kg; (**D**,**D′**) *C. orchioides* 400 mg/kg; (**E**,**E′**) Heat stress at 40 °C group; (**F**,**F′**) *C. orchioides* 100 mg/kg and heat stress; (**G**,**G′**) *C. orchioides* 200 mg/kg and heat stress; (**H**,**H′**) *C. orchioides* 400 mg/kg and heat stress; **Le**: Leydig cells; **Se**: Sertoli cells; **Sg**: Spermatogonia; **Sc**: Spermatocytes; **rs**: Round spermatids; **es**: Elongating spermatids; **S**: Spermatozoa.

Meanwhile, the heat-stress-exposed mice exhibited degenerated and disorganized features of spermatogenic epithelium and reduced spermatogenic cell numbers (Figure 3E,E′). The structural degeneration of seminiferous tubules and spermatogenesis was demonstrated by the declining size of the seminiferous tubules, the germ cells exhibiting severe sloughing and deterioration, and the appearance of abnormal spaces and giant cells in the seminiferous tubules, seminiferous tubules absent spermatozoa, and secondary sperma-

tocytes. However, testicular tissue structure and spermatogenesis in mice that received
*C. orchioides* extract, especially at doses of 200 mg/kg and 400 mg/kg, were significantly
restored. The seminiferous tubules are improved with relatively uniform tubule sizes, and
sufficient spermatogenic cells (Figure 3G,G',H,H').

Spermatogenesis was assessed using Johnsen's mean testicular biopsy score which
is shown in Table 3 and Figure 4. Johnsen's score in the control group was 8.163 ± 0.109.
Johnsen's score of mice treated with *C. orchioides* extract at doses of 100 mg/kg, 200 mg/kg,
and 400 mg/kg was not significantly different from those in the control group (8.144 ± 0.159,
8.263 ± 0.102, and 8.326 ± 0.175, respectively).

**Table 3.** Characterization of testicular histopathology according to Johnson's Scoring system.

| Groups | Treatment | Johnson's Score |
|---|---|---|
| C | Distilled water | 8.163 ± 0.109 |
| C100 | 100 mg/kg | 8.144 ± 0.159 |
| C200 | 200 mg/kg | 8.263 ± 0.102 |
| C400 | 400 mg/kg | 8.326 ± 0.175 |
| H | 40 °C + Distilled water | 5.000 ± 0.414 |
| HC100 | 40 °C + 100 mg/kg | 5.815 ± 0.191 |
| HC200 | 40 °C + 200 mg/kg | 6.300 ± 0.202 |
| HC400 | 40 °C + 400 mg/kg | 6.374 ± 0.222 |

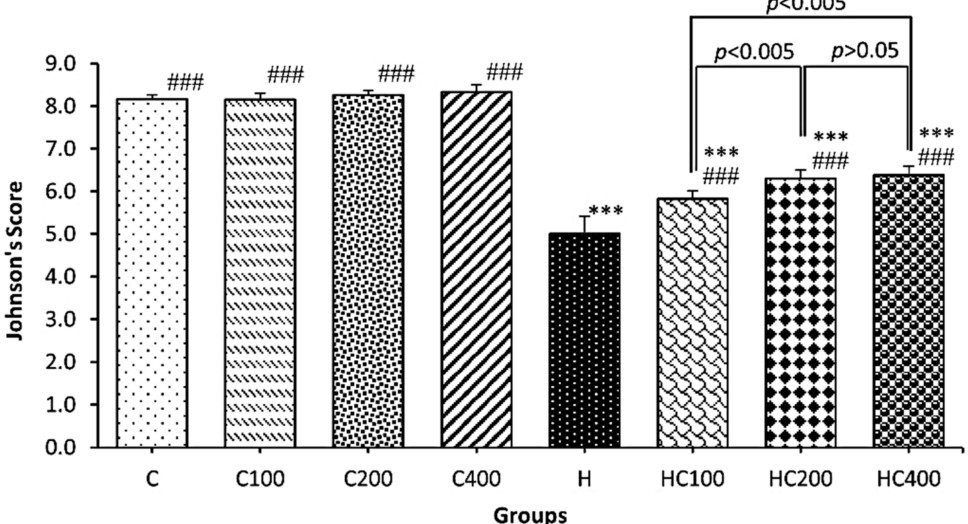

**Figure 4.** Evaluating Testicular Damage in mice subjected to heat exposure and treated with
*C. orchioides* extract by the Johnsen scoring system. * Compared to the control group, # Compared to
the heat stress groups, ***, ###: $p < 0.005$.

Johnson's score of mice subjected to 40 °C was greatly reduced (5.000 ± 0.414). However, heat-stressed male mice treated with *C. orchioides* extract showed significant recovery
in spermatogenesis. Mice exposed to 40 °C combined with treatment with *C. orchioides* extract at the doses of 100 mg/kg, 200 mg/kg, and 400 mg/kg (HC100, HC200, HC400 groups)
significantly increased Johnson scores compared to the heat-stress group (5.815 ± 0.191,
6.300 ± 0.202, and 6.374 ± 0.222, respectively). Johnson's score in heat-stress mice combined
with *C. orchioides* extract at doses of 400 mg/kg was not significantly different from that of
those in the treatment 200 mg/kg group.

The frequency distribution of Johnsen scores of seminiferous tubular cross-sections in
the eight male mice group study is shown in Figure 5. The results show that the groups
without heat stress exposure (C, C100, C200, and C400 group) could observe scores of 9–10
and scores 7–8, but never observe scores of 1–2 and scores 3–4. Scores 7–8 were accounted
for mainly in the groups without heat stress exposure (C, C100, C200, and C400 group).
The heat stress exposure group demonstrated dramatically decreased scores of 5–6, while

the combined *C. orchioides* extract ground demonstrated increased scores of 7–8. Scores of 5–6 were accounted for mainly in the heat stress exposure group and were decreased when combined with *C. orchioides* extract treatment (HC100, HC200, and HC400). Scores 1–2 appeared in the heat stress exposure group (H) and the heat stress combined 100 mg/kg extract (HC100) group, while they were almost absent in the heat stress combined 200 and 400 mg/kg extract (HC200 and HC400) groups.

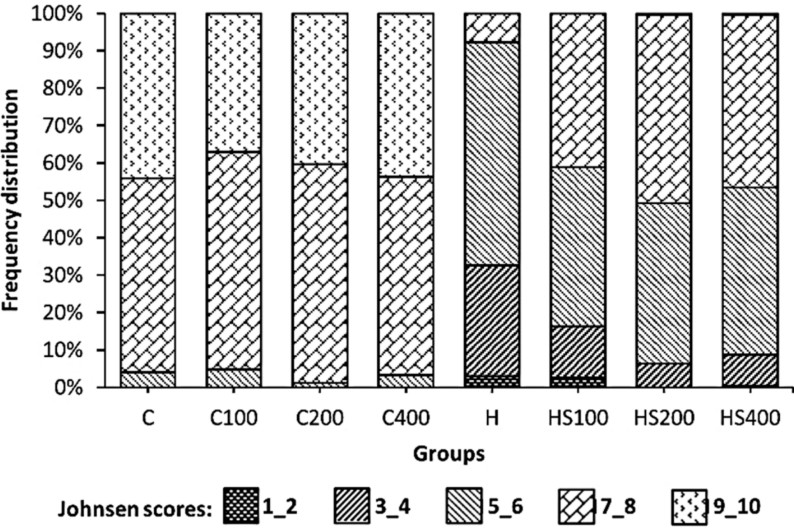

**Figure 5.** Frequency distribution of Johnsen scores of seminiferous tubular cross-sections in mice subjected to heat exposure and treated with *C. orchioides* extract.

## 4. Discussion

Spermatogenesis is a complex process involving a series of successive cell divisions in the testicles during puberty and lasting until the male's lifetime. Newly formed sperm are immobile and are released along the vas deferens to reach the epididymis. In the epididymis, sperm matures and develops motility, and then is stored in the tail of the epididymis for ejaculation [33]. Spermatogenesis in mammals is a complex process that occurs in the seminiferous tubules of the testes. Spermiogenesis is temperature-dependent, so testes in many mammalian species are extruded outside the body to maintain a temperature 2–4 °C below body temperature. The increased testicular temperature hurts mammalian spermatogenesis. Therefore, even low levels of heat stress can reduce sperm quality and increase the risk of infertility [24,34,35].

Spermatogenesis is a complex process that depends on intratesticular and extra-testicular hormones through the intermediary testosterone. Testosterone is the primary male gonadal hormone produced by the Leydig interstitial cells of the testes and is the primary male hormone responsible for regulating sexual differentiation and producing male sexual characteristics and fertility as well as spermatogenesis [36–38]. In humans, testosterone is produced mainly in the testicles (up to 95%) and to a small extent in the adrenal glands (about 4%) of men [39]. Testosterone in the blood is an important and necessary element for the maintenance of normal libido and penile erection in most men [40]. This hormone is also used to stimulate cells of the epididymis and seminal vesicles, to stimulate the development and function of the prostate and seminal vesicles, and to activate and coordinate the spermatogenesis [41,42]. A lack of testosterone leads to a loss of libido and reductions in sperm quantity and quality [43]. Qualitatively and quantitatively adequate spermatogenesis requires testosterone, so a decline in testosterone levels is the cause found in 20 to 30% of male infertility cases [44,45].

Our results showed that *C. orchioides* rhizomes extract increases the testosterone concentration in the blood of heat stress-induced mice. Chauhan et al. (2010) also showed that after taking the extract continuously for 28 days, mice had a strong increase in the concentration of sex hormones, especially total testosterone in the serum [40]. The previous

studies also demonstrated that the anabolic effect of *C. orchioides* is exhibited in a dose-dependent manner. The results showed that serum testosterone levels increased in male rats at doses of 100 mg/kg and 200 mg/kg body weight. *C. orchioides* was also most effective in enhancing sexual behavior and improving erectile function and ejaculatory/intromission frequency [46]. Moreover, Mishra et al. found that the extract of *C. orchioides* rhizomes not only increased serum testosterone but also enhanced the protection of the male reproductive organs from cadmium-induced toxicity and improved motility and sperm density [43]. Bagchi et al. (2017) showed similar results after 28 days of high-dose *C. orchioides* extract (50 mg/kg body weight), but there was no significant increase in serum-free testosterone at low doses [47].

Stress damage of testicular tissue caused by heat stress is the main cause of spermatogenesis disorder. Testicular heat stress lowered serum testosterone levels, whereas the levels of follicle-stimulating hormone and luteinizing hormone were not significantly changed. The ultrastructure of the seminiferous tubules became abnormal after testicular heat stress, the structure of the blood–testis barrier became loose, and the Sertoli cells showed a trend of differentiation [48]. Heat stress decreased epithelial thickness, the loosening and vacuolization of the germinal epithelium, the presence of cellular debris in the lumen, and the appearance of multinucleated giant cells. The majority of the seminiferous tubules contained an incomplete layer of primary spermatocytes and no spermatids [49].

Heat stress causes complete spermatogenic arrest or disrupts spermatogenesis by activating several cellular signaling pathways such as apoptosis signaling pathways and non-apoptotic signaling pathways, as well as reactive oxygen species (ROS) production in testicular tissue. Scrotal hyperthermia significantly causes complete spermatogenic arrest with increased reactive oxygen species (ROS) production, mitochondrial membrane permeability (MMP), and oxidized glutathione (GSSG). [50]. Heat stress increased caspase 3/7 activity, protein carbonyl (PC, a measure of ROS) contents, and nitrates/nitrites (NOx, a metabolite of nitric oxide) levels in testicular tissues, which led to enhanced apoptosis. Therefore, elevated temperature drastically increases oxidative stress and cellular apoptosis, which, in turn, leads to decreased testicular functions in oysters [51]. Hasani et al. showed that heat stress disrupts spermatogenesis by activating several non-apoptotic signaling pathways in testicular tissue through the protein expression of Caspase-1, Beclin1, Atg7, MLKL, and Acsl4, together with the expression of Caspase-1, Beclin1, Atg7, MLKL, and Acsl4 [52].

Heat stress also has a destructive effect on spermatogenesis and reduces sperm quality. Analyzing the testicular tissues showed a marked decrease in sperm parameters and serum testosterone level in mice induced by scrotal hyperthermia, and also indicated a significant reduction in testicular cells and changes in the spatial arrangement of testicular cells, which finally influences the normal function of spermatogenesis [53]. On the other hand, heat stress disrupts spermatogenesis by activating several non-apoptotic signaling pathways in testicular tissue. [52]

In conclusion, *C. orchioides*, also known as black gold, is a source of bioactive compounds that has drawn growing attention for its potentially beneficial effects in preventing and treating several diseases. Our finding indicated that *C. orchioides* extract could improve spermatogenesis and testosterone levels in scrotal hyperthermia-induced mice. *C. orchioides* extract could be a potential alternative treatment for improving spermatogenesis and male reproduction. Further studies are needed to better clarify the signaling pathway of the protective effects of *C. orchioides* on spermatogenesis.

**Author Contributions:** Conceptualization, T.N.-T. and T.-N.B.-L.; methodology, T.-N.B.-L., Q.H.-T., H.H.-V., B.-P.T.-T. and T.N.-T.; investigation, T.-N.B.-L., H.H.-V. and T.N.-T.; data curation, T.-N.B.-L. and T.N.-T.; writing original draft preparation, T.-N.B.-L. and T.N.-T.; writing review and editing, T.-N.B.-L. and T.N.-T.; funding acquisition, T.N.-T. and T.-N.B.-L. All authors have read and agreed to the published version of the manuscript.

**Funding:** This study was financially supported by the Hue University-level research projects in science and technology, Hue, Vietnam (DHH 2020-04-125).

**Institutional Review Board Statement:** All animal experimentation studies were approved by the Animal Ethics Committee of Hue University, Vietnam (Certificate no. HU VN0007).

**Informed Consent Statement:** Not applicable.

**Data Availability Statement:** Data available within the article.

**Acknowledgments:** The authors also acknowledge the partial support of Hue University under the Core Research Program, The Research Group on Regenerative Medicine (NCM.DHH.2022.02).

**Conflicts of Interest:** The authors declare no conflict of interest.

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
