# Peer review of "Protective Effect of Curculigo orchioides Gaertn. Extract on Heat Stress-Induced Spermatogenesis Complications in Murine Model"

_cimb, doi:10.3390/cimb45040212_

Round 1
Reviewer 1 Report
Comments and Suggestions for Authors
Protective Effect of Curculigo Orchioides Gaertn. Extract on 3 Heat Stress-Induced Spermatogenesis Complications in Murine Model
Authors work is appreciated but extensive revision is required.
Q1. Line 2 heading genus name should start with capital letter and species name should start with small letter. Curculigo orchioides
Q2. Line 13 of Abstract; “Gaertn” should be “Gaertn.”
Q3. Line 28 and 110, since the name of herbs has genus and species it should be written in italics. Curculigo orchioides.
Q4. Line 47, the name of herbs is not in italics.
Q5. Mice number is controversial. According to Abstract and line 120-121 there should be 48 mice but line 114 mentioned about 40 mice only.
Q6. Make proper order of groups used in experiment. Groups divided in Abstact and line 120 to 125 is not even. Groups should match with your actual findings.
Q7. Line beginning with 131 ending at 133 should be removed. It seems as repetition. “The four groups were kept in different cages and had free access to water and food. All animals were cared for under identical environmental conditions.”
Q8. Write catalogue# of kit where required.
Q9. Line 148, “At the end of the surgical procedure”, remove this line and use After Euthanasia.
Q10. Line 152, is it tap water or distilled water for rehydration?
Q11. Line 166, letter “A” should be capital after word Student’s t-test.
Q12. Line 168, word should be “Results”
Q12. Remove Figure 2, blood testosterone level. Table looks fine for it. Statistical marker should be put in proper order. ### mark does not match with Figure legends.
Q13. Similar revision is suggested for Figure 4. As Q12. Table looks fine for it. Statistical marker should be put in proper order. ### mark does not match with Figure legends.
Q14. Line 244, please see the grammar marks, 5.000 is written as 5,000.
Q15.Line 302, it should be 28 days not 28 day.
Q16. Show IHC or Western blot for ROS production due to heat stress, writing in only discussion would not support your comments. Also show for Caspase.
Q17. Line 318, Heat is written as Head
Author Response
Reviewer 1:
Authors work is appreciated but extensive revision is required.
Q1. Line 2 heading genus name should start with capital letter and species name should start with small letter. Curculigo orchioides
Response: Thank you for pointing out a mistake. We have corrected it according to your comment
Q2. Line 13 of Abstract; “Gaertn” should be “Gaertn.”
Response: Thank you for pointing out a mistake. We have corrected it according to your comment
Q3. Line 28 and 110, since the name of herbs has genus and species it should be written in italics. Curculigo orchioides.
Response: Thank you for pointing out a mistake. We have corrected it according to your comment
Q4. Line 47, the name of herbs is not in italics.
Response: Thank you for pointing out a mistake. We have corrected it according to your comment
Q5. Mice number is controversial. According to Abstract and line 120-121 there should be 48 mice but line 114 mentioned about 40 mice only.
Response: Thank you for pointing out a mistake. We have corrected the number of mice to 48
Q6. Make proper order of groups used in experiment. Groups divided in Abstact and line 120 to 125 is not even. Groups should match with your actual findings.
Response: Thank you for pointing out a mistake. We have corrected it according to your comment
Q7. Line beginning with 131 ending at 133 should be removed. It seems as repetition. “The four groups were kept in different cages and had free access to water and food. All animals were cared for under identical environmental conditions.”
Response: Thank you for pointing out a mistake. We have corrected it according to your comment
Q8. Write catalogue# of kit where required.
Response: Thank you for pointing out a mistake. We have corrected it according to your comment
Q9. Line 148, “At the end of the surgical procedure”, remove this line and use After Euthanasia.
Response: Thank you for pointing out a mistake. We have corrected it according to your comment
Q10. Line 152, is it tap water or distilled water for rehydration?
Response: Thank you for your comment! We use distilled water for rehydrated step.
Q11. Line 166, letter “A” should be capital after word Student’s t-test.
Response: Thank you for pointing out a mistake. We have corrected it according to your comment
Q12. Line 168, word should be “Results”
Response: Thank you for pointing out a mistake. We have corrected it according to your comment
Q12. Remove Figure 2, blood testosterone level. Table looks fine for it. Statistical marker should be put in proper order. ### mark does not match with Figure legends.
Response: Thank you for your comment. We would like to present the blood testosterone level in Table 2 to show each group's mean and SD values. Statistical analysis to show the mean difference between the group was shown in Figure 2. We corrected the statistical marker in Figure 2.
Q13. Similar revision is suggested for Figure 4. As Q12. Table looks fine for it. Statistical marker should be put in proper order. ### mark does not match with Figure legends.
Q14. Line 244, please see the grammar marks, 5.000 is written as 5,000.
Response: Thank you for pointing out a mistake. We have corrected it according to your comment
Q15.Line 302, it should be 28 days not 28 day.
Response: Thank you for pointing out a mistake. We have corrected it according to your comment
Q16. Show IHC or Western blot for ROS production due to heat stress, writing in only discussion would not support your comments. Also show for Caspase.
Response: Thank you for your comment! We discussed some molecular signaling as published in [52] [53][54].
Q17. Line 318, Heat is written as Head
Response: Thank you for pointing out a mistake. We have corrected it according to your comment
Reviewer 2 Report
Comments and Suggestions for Authors
Article: Protective Effect of Curculigo Orchioides Gaertn. Extract on Heat Stress-Induced Spermatogenesis Complications in Murine Model
Summary
Authors present a study that shows a plant extract commonly used in Asian countries may protect male mouse gonads from heat stress damage. The protective effect was assessed by measuring testosterone levels and via testicular histopathology. The study is descriptive in nature and doesn't offer any possible mechanism for this protective effect.
Questions and comments:
Line 13: Abstract should also include observation as to what effect C. orchioides had on animals without heat stress. i.e. C. orchioides did not significantly change testosterone levels or testicular histopathology in non-heat treated animals
Line 54: "Male infertility has been reported to be a major cause of infertility in at least 50% of all infertile couples [24]."
Comment: strange that authors would cite a small rat study for evidence of world-wide human male infertility. A better study to cite could be something like (PMID: 36377604)
Line 70: How many C. orchioides rhizomes from how many plants were harvested in total for processing?
Line 71: "Plants were cleaned, dried, and pulverized " would be better if this process was better explained just in case processing influenced the herbal powder generated
Line 73: "ultrasonicated"
What sonicator settings were used? what machine was used?
Line 74: "stored it in a refrigerator at 4°C for further use"
Comment: how long was it stored in general? days, weeks, months, years? Were experiments done using extract stored for the same length of time?
Line 77: "The extract was concentrated to get a brownish mass"
Comment: How was it concentrated?
Line 78: "diluted with water"
Comment: what kind of water?
Line 79: "administered orally as a suspension using a metal cannula"
Comment: do you mean gavaged? was a gavage cannula used?
Comment: This information should be moved to "animals" section
Line 81: "using Mayer’s, Dragendorff’s, and Bouchardat’s reagents"
Comment: were these reagents made by authors, or purchased?
Line 98: "HPLC system of Thermo Electron"
What was model number of equipment used?
Line 99: "ChromQuest software"
What software version was used?
Line 108: Figure 1.
Comment: figure 1 doesn't really highlight any significant data for reader. At best, should only be included as supplemental material.
Line 127: "For five weeks, the lower half of the mouse's body, including the scrotum, was immersed in a temperature-controlled water bath (40 °C) for 10 minutes twice per day at 10-minute intervals"
Comment: how did you keep only the lower half of the mouse body in the water bath for 10 minutes? If anaesthesia was used, please outline.
Line 120: Question: When did C. orchioides extract dosing begin? the same day as water bath treatment? or during the 7 day acclimation period?
Line 134: "were sacrificed under anesthesia."
What anesthesia was used? how was it given, how was euthanasia confirmed?
Line 138: "The serum concentration of total testosterone was measured using the electrochemi luminescence immunoassay (ECLIA) kit"
Comment: please provide kit number, and information on equipment used to read kit
Line 148: "immersed into 4% buffered formaldehyde"
Question: how big were the samples, and how long were samples fixed in formaldehyde for?
Line 150: "Paraffin sections 5 μm thick were obtained "
Question: how were these obtained, what equipment was used?
Line 153: "hematoxylin and eosin"
Question: where were these chemicals obtained?
Line 158: "All tubular sections in one section of the testicular biopsy are systematically and each is given a score from 1 to 10 according to the following Johnsen scoring system"
Question: who did the scoring? (i.e someone with experience using Johnsen scoring system?), where these scored blindly?
Line 165: "The significant differences among the means of the treated groups were analyzed using a Student’s t-test."
Comment: students t-test seems a little basic, why wasn't at least ANOVA used?
Line 182: "Quantitative analysis of the testosterone levels in the blood of mice subjected to heat stress with or without C. orchioides extract treatment after 6 weeks is presented in Table 2 and Figure 2"
Question: why does text and tables say testosterone was measured in blood, but methods list testosterone as being measured from homogenized testicular samples (Line 140)?? These are not the same.
Lines 204, 234: Tables 2 and 3.
Comment: I assume H-group was also given distilled water to drink, not just C-group? Tables may confuse reader.
Line 396: Reference 23 (International Journal of Applied Research In Natural Products?) is a predatory journal and may not be adequately peer-reviewed
Suggestions
Line 40: suggest change "C. orchioides has been widely used in Asian indigenous medicine systems to treat a variety of diseases including diabetes [12,13], hepatoprotective, antioxidant, and anti-cancer [14,15], protect the nervous system, enhance memory [16], improve the immunity system and rejuvenation [5,17], anti-inflammatory [12,14]." to
"C. orchioides has been widely used in Asian indigenous medicine and has suggested hepatoprotective, antioxidant, anti-cancer [14,15], neuroprotecive, and anti-inflammatory affects [12,14]. C. orchioides has also been used to enhance memory [16], improve the immune system and rejuvenation [5,17], and to treat a variety of diseases including diabetes [12,13]."
Line 46: suggest change "purified the active substances in the crude polysaccharide from C. orchioides and show that it significantly enhanced the mineralization of the osteoblasts"
"purified two polysaccharides from C. orchioides and showed they significantly enhanced the mineralization of osteoblasts"
Line 49: suggest change "The evidence from an animal experiment shows" to "Chauhan et al. showed"
Line 68: suggest change "Material and method" to "Materials and methods"
Line 70: suggest change "The rhizomes of C. orchioideswere harvested from Thua Thien Hue province" to "Rhizomes of C. orchioides were harvested in the Thua Thien Hue province"
Line 73: suggest change "Filter the extract with a 0.45 μm Minissart filter" to "The extract was subsequently filtered using a 0.45 μm Minissart filter"
Line 121: suggest change "(C) Control group just drink distilled water; (H) 40°C heat exposure group; (C100) C. orchioides extract 100 mg/kg/day group; (C200) 200 mg/kg/day C. orchioides extract group; (C400) 400 mg/kg/day C. orchioides extract group; (HC100) 100 mg/kg/day C. orchioides extract + heat stress group; (HC200) 200 mg/kg/day C. orchioides extract + heat stress group; (HC400) 400 mg/kg/day C. orchioides extract + heat stress group." to
"(C) Control group and (H) 40°C heat exposure groups were both given plain distilled water to drink; (C100) C. orchioides extract 100 mg/kg/day group; (C200) 200 mg/kg/day C. orchioides extract group; (C400) 400 mg/kg/day C. orchioides extract group; (HC100) 100 mg/kg/day C. orchioides extract + heat stress group; (HC200) 200 mg/kg/day C. orchioides extract + heat stress group; (HC400) 400 mg/kg/day C. orchioides extract + heat stress group."
Line 148: suggest change "the testicular specimens" to "testicular specimens"
Line 170: suggest change "The presence of alkaloids was evidenced" to "The presence of alkaloids in C. orchioides extracts was evidenced"
Author Response
Reviewer 2:
Summary
Authors present a study that shows a plant extract commonly used in Asian countries may protect male mouse gonads from heat stress damage. The protective effect was assessed by measuring testosterone levels and via testicular histopathology. The study is descriptive in nature and doesn't offer any possible mechanism for this protective effect.
Response: Thank you very much for your time, consideration as well as your useful comments.
Questions and comments:
Line 13: Abstract should also include observation as to what effect C. orchioides had on animals without heat stress. i.e. C. orchioides did not significantly change testosterone levels or testicular histopathology in non-heat treated animals
Response: Thank you for your comment! We have added the description of the control group without heat stress in abstract
Line 54: "Male infertility has been reported to be a major cause of infertility in at least 50% of all infertile couples [24]."
Comment: strange that authors would cite a small rat study for evidence of world-wide human male infertility. A better study to cite could be something like (PMID: 36377604)
Response: Thank you for your comment! We have revised this sentence and changed citation (PMID: 36377604)
Line 70: How many C. orchioides rhizomes from how many plants were harvested in total for processing?
Line 71: "Plants were cleaned, dried, and pulverized " would be better if this process was better explained just in case processing influenced the herbal powder generated
Line 73: "ultrasonicated"
What sonicator settings were used? what machine was used?
Response: Thank you for your comment! We have revised "Plant collection and extraction" per your comments
Line 74: "stored it in a refrigerator at 4°C for further use"
Comment: how long was it stored in general? days, weeks, months, years? Were experiments done using extract stored for the same length of time?
Response: Thank you for your comment! The extract used in this experiment was prepared at the same time. The extract was stored at 4°C and used to treat mice for a month.
Line 77: "The extract was concentrated to get a brownish mass"
Comment: How was it concentrated?
Response: Thank you for your comment! The extract was concentrated at 45 °C to get a brownish mass
Line 78: "diluted with water"
Comment: what kind of water?
Response: Thank you for your comment! The extract was diluted with distilled water
Line 79: "administered orally as a suspension using a metal cannula"
Comment: do you mean gavaged? was a gavage cannula used?
Comment: This information should be moved to "animals" section
Response: Thank you for your comment!
Aqueous extracts were administered orally as a suspension using 20-gauge stainless steel feeding needles. We have revised and moved this sentence to the animal section.
Line 81: "using Mayer’s, Dragendorff’s, and Bouchardat’s reagents"
Comment: were these reagents made by authors, or purchased?
Response: Thank you for your comment! We purchased their reagents from the company. We added the reagents information to the manuscript.
Line 98: "HPLC system of Thermo Electron"
What was model number of equipment used?
Response: Thank you for your comment!
HPLC analysis was carried out using an Accela HPLC liquid phase system (Thermo Fisher Scientific, Waltham, MA, USA).
Line 108: Figure 1.
Comment: figure 1 doesn't really highlight any significant data for reader. At best, should only be included as supplemental material.
Response: Thank you for your comment! We would like to describe the characteristics of herbal material from the whole plant in nature, the rhizomes, to the quantification of Lycorine in Curculigo orchioides in Figure 1.
Line 127: "For five weeks, the lower half of the mouse's body, including the scrotum, was immersed in a temperature-controlled water bath (40 °C) for 10 minutes twice per day at 10-minute intervals"
Comment: how did you keep only the lower half of the mouse body in the water bath for 10 minutes? If anaesthesia was used, please outline.
Response: Thank you for your comment!
The lower half of the mouse's body was immersed in a temperature-controlled water bath (40°C) using a keeper tool for 10 minutes twice per day at 10-minute intervals
Line 120: Question: When did C. orchioides extract dosing begin? the same day as water bath treatment? or during the 7 day acclimation period?
Response: Thank you for your comment!
- orchioides extract was prepared one week before the acclimation period
Line 134: "were sacrificed under anesthesia."
What anesthesia was used? how was it given, how was euthanasia confirmed?
Response: Thank you for your comment!
After completing the heat exposure experiment for 5 weeks, mice were anesthetized with 80 mg/kg ketamine and 16 mg/kg xylazine and then performed cardiac puncture for blood collection as a terminal procedure.
Line 138: "The serum concentration of total testosterone was measured using the electrochemi luminescence immunoassay (ECLIA) kit"
Comment: please provide kit number, and information on equipment used to read kit
Response: Thank you for your comment!
The serum concentration of total testosterone was measured by electrochemiluminescence immunoassay using Elecsys Testosterone II (05200067190, Roche, Basel, Switzerland) from Roche Diagnostics according to the manufacturer’s instructions with the cobas ® 8000 modular (Roche Diagnostics, Mannheim, Germany).
Line 148: "immersed into 4% buffered formaldehyde"
Question: how big were the samples, and how long were samples fixed in formaldehyde for?
Response: Thank you for your comment!
The testicular specimens were individually immersed in 4% buffered formaldehyde for 24 hours. (whole testis tissue around 3x5mm)
Line 150: "Paraffin sections 5 μm thick were obtained "
Question: how were these obtained, what equipment was used?
Response: Thank you for your comment!
Paraffin sections 5 μm thick were obtained by microtome (Leica RM2255, Germany)
Line 153: "hematoxylin and eosin"
Question: where were these chemicals obtained?
Response: Thank you for your comment!
Slides were stained with hematoxylin and eosin (H&E) (Sigma, Vienna, Austria) and observed under a light microscope for histopathological analysis.
Line 158: "All tubular sections in one section of the testicular biopsy are systematically and each is given a score from 1 to 10 according to the following Johnsen scoring system"
Question: who did the scoring? (i.e someone with experience using Johnsen scoring system?), where these scored blindly?
Response: Thank you for your comment!
Testicular biopsies were blinded and given a score according to the following Johnsen scoring system by two independent author
Line 165: "The significant differences among the means of the treated groups were analyzed using a Student’s t-test."
Comment: students t-test seems a little basic, why wasn't at least ANOVA used?
Response: Thank you for your comment!
We used the Student's t test to compare the means between two groups
Line 182: "Quantitative analysis of the testosterone levels in the blood of mice subjected to heat stress with or without C. orchioides extract treatment after 6 weeks is presented in Table 2 and Figure 2"
Question: why does text and tables say testosterone was measured in blood, but methods list testosterone as being measured from homogenized testicular samples (Line 140)?? These are not the same.
Response: Thank you for pointing out a mistake.
The testosterone level was measured in the circulation blood collected by cardiac puncture. We have corrected the Method.
Mice were anesthetized and then collected the circulation blood by cardiac puncture. The blood serum concentration of total testosterone was measured by electrochemiluminescence immunoassay using Elecsys Testosterone II (05200067190, Roche, Basel, Switzerland) from Roche Diagnostics according to the manufacturer’s instructions with the cobas ® 8000 modular (Roche Diagnostics, Mannheim, Germany). The sensitivities of hormone detected per assay tube were 0.025ng/mL.
Lines 204, 234: Tables 2 and 3.
Comment: I assume H-group was also given distilled water to drink, not just C-group? Tables may confuse reader.
Response: Thank you for your comment! We have revised "40°C + Distilled water" in Table 2 and 3 according to your comments
Line 396: Reference 23 (International Journal of Applied Research In Natural Products?) is a predatory journal and may not be adequately peer-reviewed
Suggestions
Response: Thank you for your comment! We cited this paper that was published in "International Journal of Applied Research In Natural Products" in 2008. We have checked in scimago website and saw that this journal was listed in Scopus from 2008 to 2016 (https://www.scimagojr.com/journalsearch.php?q=19900193502&tip=sid&clean=0)
Line 40: suggest change
"C. orchioides has been widely used in Asian indigenous medicine systems to treat a variety of diseases including diabetes [12,13], hepatoprotective, antioxidant, and anti-cancer [14,15], protect the nervous system, enhance memory [16], improve the immunity system and rejuvenation [5,17], anti-inflammatory [12,14]." to
"C. orchioides has been widely used in Asian indigenous medicine and has suggested hepatoprotective, antioxidant, anti-cancer [14,15], neuroprotecive, and anti-inflammatory affects [12,14]. C. orchioides has also been used to enhance memory [16], improve the immune system and rejuvenation [5,17], and to treat a variety of diseases including diabetes [12,13]."
Response: Thank you for your comment! We have revised it according to your comments
Line 46: suggest change "purified the active substances in the crude polysaccharide from C. orchioides and show that it significantly enhanced the mineralization of the osteoblasts"
"purified two polysaccharides from C. orchioides and showed they significantly enhanced the mineralization of osteoblasts"
Response: Thank you for your comment! We have revised it according to your comments
Line 49: suggest change "The evidence from an animal experiment shows" to "Chauhan et al. showed"
Response: Thank you for your comment! We have revised it according to your comments
Line 68: suggest change "Material and method" to "Materials and methods"
Response: Thank you for your comment! We have revised it according to your comments
Line 70: suggest change "The rhizomes of C. orchioideswere harvested from Thua Thien Hue province" to "Rhizomes of C. orchioides were harvested in the Thua Thien Hue province"
Response: Thank you for your comment! We have revised it according to your comments
Line 73: suggest change "Filter the extract with a 0.45 μm Minissart filter" to "The extract was subsequently filtered using a 0.45 μm Minissart filter"
Response: Thank you for your comment! We have revised it according to your comments
Line 121: suggest change
"(C) Control group just drink distilled water; (H) 40°C heat exposure group; (C100) C. orchioides extract 100 mg/kg/day group; (C200) 200 mg/kg/day C. orchioides extract group; (C400) 400 mg/kg/day C. orchioides extract group; (HC100) 100 mg/kg/day C. orchioides extract + heat stress group; (HC200) 200 mg/kg/day C. orchioides extract + heat stress group; (HC400) 400 mg/kg/day C. orchioides extract + heat stress group." to
"(C) Control group and (H) 40°C heat exposure groups were both given plain distilled water to drink; (C100) C. orchioides extract 100 mg/kg/day group; (C200) 200 mg/kg/day C. orchioides extract group; (C400) 400 mg/kg/day C. orchioides extract group; (HC100) 100 mg/kg/day C. orchioides extract + heat stress group; (HC200) 200 mg/kg/day C. orchioides extract + heat stress group; (HC400) 400 mg/kg/day C. orchioides extract + heat stress group."
Response: Thank you for your comment! We have revised it according to your comments
Line 148: suggest change "the testicular specimens" to "testicular specimens"
Response: Thank you for your comment! We have revised it according to your comments
Line 170: suggest change "The presence of alkaloids was evidenced" to "The presence of alkaloids in C. orchioides extracts was evidenced"
Response: Thank you for your comment! We have revised it according to your comments
Reviewer 3 Report
Comments and Suggestions for Authors
Hypethermia and its effects on the testicular function has become a very intriguing topic in the field of andrology, and will become a research highlight as an accompanying phenomenon of global warming. As such, any novel strategies for the prevention or management of male reproductive dysfunction as a result of high temperatures. The study present with the potential to attract the interest of readers and with novel data. Nevertheless, there are several aspects that could be improved:
- The topic of hyperthermia should be extended a bit in the Introduction section. What are the consequences of testicular hypethermia?
- Why only testosterone was measured? Data on the endocrine profile could have been supported by measurements of FSH andLH on the systemic level or perhaps by DHEA, cholesterol or androstenedione to unravel possible changes in the steroid synthesis.
- Information on oxidative stress as a hallmark of hyperthermia-induced infertility can be omitted since no oxidative markers were evaluated and the antioxidant activity of the extract was not determined. I would rather focus on the outcomes supported by the collected data. All in all, the discussion should be focused more on the actual novelty - beneficial effects of the extract. What would be its mechanisms of action in relation to this paper?
- Limitations and future prospects of the study should be properly discussed.
Author Response
Reviewer 3:
Hypethermia and its effects on the testicular function has become a very intriguing topic in the field of andrology, and will become a research highlight as an accompanying phenomenon of global warming. As such, any novel strategies for the prevention or management of male reproductive dysfunction as a result of high temperatures. The study present with the potential to attract the interest of readers and with novel data. Nevertheless, there are several aspects that could be improved:
- The topic of hyperthermia should be extended a bit in the Introduction section. What are the consequences of testicular hypethermia?
Response: Thank you for your comment! We have talked about the effect of heat stress on spermatogenesis in the Introduction according to your comments
There are many factors that cause a decline in male reproductive health and semen quality [23]. Heat stress is one of the risk factors that affect most male reproductive functions in mammals. High scrotal temperature is a significant cause of male factor infertility. There are many common thermogenic factors that affect scrotal temperature, including lifestyle and behavioral influences, occupational and environmental factors, and pathophysiological causes [24]. Spermatogenesis is a temperature-dependent process, and any increase above the normal scrotal temperature range can result in the death of heat-vulnerable germ cells and disrupted somatic cell function [25].
- Why only testosterone was measured? Data on the endocrine profile could have been supported by measurements of FSH and LH on the systemic level or perhaps by DHEA, cholesterol or androstenedione to unravel possible changes in the steroid synthesis.
Response: Thank you for your comment!
Testosterone is the primary male hormone responsible for regulating sex differentiation, producing male sex characteristics, spermatogenesis, and fertility. We measure blood testosterone levels to examine the effect of the herbal extract on the male reproduction system. In near future, we will also check other hormones such as FSH and LH
- Information on oxidative stress as a hallmark of hyperthermia-induced infertility can be omitted since no oxidative markers were evaluated and the antioxidant activity of the extract was not determined. I would rather focus on the outcomes supported by the collected data. All in all, the discussion should be focused more on the actual novelty - beneficial effects of the extract. What would be its mechanisms of action in relation to this paper?
Response: Thank you for your comment! We have revised the Discussion according to your comments
Stress damage of testicular tissue caused by heat stress is the main cause of spermatogenesis disorder. Testicular heat stress lowered serum testosterone levels, whereas the levels of follicle-stimulating hormone and luteinizing hormone were not significantly changed. The ultrastructure of the seminiferous tubules became abnormal after testicular heat stress, the structure of the blood-testis barrier became loose, and the Sertoli cells showed a trend of differentiation [48]. Heat stress decrease in the epithelial thickness, loosening, and vacuolization of the germinal epithelium, presence of cellular debris in the lumen, the appearance of multinucleated giant cells, and pyknotic cells with a fragmented nucleus, majority of the seminiferous tubules contained spermatogonia with or without an incomplete layer of primary spermatocytes and no spermatids [49]. Heat stress causes complete spermatogenic arrested or disrupts spermatogenesis by activating several cellular signalings pathways such as apoptosis signaling pathways, non-apoptotic signaling pathways, and reactive oxygen species (ROS) production in testicular tissue [50][51][52].
- Limitations and future prospects of the study should be properly discussed.
Response: Thank you for your comment! We have revised the Discussion according to your comments. Further studies are needed to better clarify the signaling pathway of the protective effects of C. orchioides on spermatogenesis
Round 2
Reviewer 1 Report
Comments and Suggestions for Authors
Authors have revised minor changes that is suggested but have ignored major changes
Q1. According to Review 1. ### is not required in all the groups. Why have authors put statistical mark from control to last group? Statistical mark should only be in H and H+treatment groups. Authors were suggested to remove some figures, tables are enough for representing your data.
Q2.Removing oxidative stress from discussion is not accepted because your data is small and only one IHC figure will not fit for the standard of IJMS. So Authors are suggested to work on oxidative stress markers and other IHC or Western blot findings whatever to elaborate your results. If you are not addressing it. It would not support what may be reason that Heat is causing infertility. These findings will show mechanism of action.
Q3.With these small minor changes I would not support for IJMS.
Author Response
Q1. According to Review 1. ### is not required in all the groups. Why have authors put statistical mark from control to last group? Statistical mark should only be in H and H+treatment groups. Authors were suggested to remove some figures, tables are enough for representing your data.
Response:
Thank you for your comment!
In order to compare without heat stress groups (C group, C100 group, C200 group, and C400 group), the heat stress combined with C. orchioides extract treatment groups (HC100, HC200, and HC400) and the heat stress only group, we put statistical mark # to compare to the heat stress group.
Results in Figure 2 shows that testosterone level in the heat stress group significantly decreased compared to those in the without heat stress group (C group, C100 group, C200 group, and C400 group) (###, p<0.005). In the heat stress combined with C. orchioides extract treatment groups, just treatment combine 200 and 400 mg/kg (HC200, HC400 group) significantly increase the testosterone levels compared to the heat stress (##, p<0.05). Testosterone levels in the heat stress combined 100 mg/kg (HC100) were not significantly different from those in the heat stress group.
Results in Figure 2 show that Johnsen score in both the without heat stress group (C group, C100 group, C200 group, and C400 group) and the heat stress combined with C. orchioides extract treatment groups (HC100, HC200, HC400 group) significantly higher compared to those in the heat stress group (H group) (###, p<0.005)
Figure 2 presents the chart of the data in table 2 with statistical analysis. Figure 4 presents the chart of the data in table 3 with statistical analysis.
Q2.Removing oxidative stress from discussion is not accepted because your data is small and only one IHC figure will not fit for the standard of IJMS. So Authors are suggested to work on oxidative stress markers and other IHC or Western blot findings whatever to elaborate your results. If you are not addressing it. It would not support what may be reason that Heat is causing infertility. These findings will show mechanism of action.
Response:
Thank you for your comment!
We have revised the Discussion according to your comment
One of our research limitations is that does not show the signaling pathways protection effect of C. orchioides extract on spermatogenesis. We show the evidence in the testosterone level in blood and histopathology findings.